# A Probiotic *Lactobacillus gasseri* Alleviates *Escherichia coli*-Induced Cognitive Impairment and Depression in Mice by Regulating IL-1β Expression and Gut Microbiota

**DOI:** 10.3390/nu12113441

**Published:** 2020-11-10

**Authors:** Soo-Won Yun, Jeon-Kyung Kim, Kyung-Eon Lee, Young Joon Oh, Hak-Jong Choi, Myung Joo Han, Dong-Hyun Kim

**Affiliations:** 1Department of Food and Nutrition, Kyung Hee University, Seoul 02447, Korea; ysw6923@naver.com (S.-W.Y.); mjhan@khu.ac.kr (M.J.H.); 2Neurobiota Research Center, College of Pharmacy, Kyung Hee University, Seoul 02447, Korea; kim_jk0225@naver.com (J.-K.K.); mljun@khu.ac.kr (K.-E.L.); 3Division of Research and Development, World Institute of Kimchi, Gwangju 61755, Korea; yjoh@wikim.re.kr (Y.J.O.); hjchoi@wikim.re.kr (H.-J.C.)

**Keywords:** *Lactobacillus gasseri*, cognition impairment, depression, IL-1β, gut microbiota

## Abstract

Excessive expression of interleukin (IL)-1β in the brain causes depression and cognitive dysfunction. Herein, we investigated the effect of *Lactobacillus gasseri* NK109, which suppressed IL-1β expression in activated macrophages, on *Escherichia coli* K1-induced cognitive impairment and depression in mice. Germ-free and specific pathogen-free mice with neuropsychiatric disorders were prepared by oral gavage of K1. NK109 alleviated K1-induced cognition-impaired and depressive behaviors, decreased the expression of IL-1β and populations of NF-κB^+^/Iba1^+^ and IL-1R^+^ cells, and increased the K1-suppressed population of BDNF^+^/NeuN^+^ cells in the hippocampus. However, its effects were partially attenuated by celiac vagotomy. NK109 treatment mitigated K1-induced colitis and gut dysbiosis. Tyndallized NK109, even if lysed, alleviated cognitive impairment and depression. In conclusion, NK109 alleviated neuropsychiatric disorders and colitis by modulating IL-1β expression, gut microbiota, and vagus nerve-mediated gut–brain signaling.

## 1. Introduction

The brain bidirectionally communicates with the gut microbiota/gastrointestinal tract via the hypothalamus–pituitary–adrenal (HPA) and microbiota–gut–brain (MGB) axes [1,2]. Excessive exposure to stresses such as social defeat, pathogens, and immobilization enhances the secretion of adrenal hormones, such as cortisol, and immune cytokines, such as interleukin (IL)-1β and IL-6 through activation of the HPA axis, resulting in gut microbiota alteration, depression, and cognitive impairment [3,4]. The overgrowth of endogenous or exogenous *Escherichia coli* in the intestine by stressors such as antibiotics causes gut microbiota alteration and collapses gut immune homeostasis, resulting in the occurrence of gut inflammation, depression, and cognitive impairment through the induction of IL-1β and corticosterone production [5,6]. Excessive IL-1β expression in the hippocampus causes psychiatric disorders such as depression [7,8]. Therefore, suppressing IL-1β expression may be beneficial for the therapy of neuropsychiatric disorders.

Probiotics are live microbes that promote health benefits [9]. Of these, Lactobacilli are repeatedly found in yogurt, kimchi, cheese, and mammalian gut bacteria. They revive altered microbiota [10,11], regulate the immune system of hosts [12], increase the efficacy of *Helicobacter pylori* eradication treatment [13], improve colitis [14], and alleviate cognitive impairment and depression [15,16]. *Lactobacillus plantarum* ATCC 8014 improves depression in streptozotocin/high-fat diet-treated mice [17]. *Lactobacillus helveticus* NS8 alleviates anxiety and cognitive impairment in IL-10-deficient mice [18]. *Lactobacillus sakei* OK67 mitigates anxiety in high-fat diet-treated mice by suppressing gut microbiota-involved NF-κB activation [14]. *Lactobacillus reuteri* NK33 and *Bifidobacterium adolescentis* NK98 mitigate anxiety, depression, and colitis by easing altered gut microbiota [16]. *Bifidobacterium longum* NK46 alleviates cognitive impairment in 5XFAD-transgenic mice by suppressing gut microbiota alteration and bacterial endotoxin production [19]. However, whether IL-1β expression-suppressing probiotics can simultaneously alleviate cognitive impairment and depression remains unclear.

Therefore, we isolated IL-1β expression-inhibitory *Lactobacillus gasseri* NK109 from the bacterial collection of human feces and examined the effects of NK109 and its lysates (membrane and cytosolic fractions) on *Escherichia coli* K1-induced gut microbiota alteration, colitis, depression, and cognitive impairment in germ-free and specific pathogen-free (SPF) mice.

## 2. Materials and Methods

### 2.1. Materials

Sodium thioglycolate, 4′,6-diamidino-2-phenylindole dilactate (DAPI), lipopolysaccharide (LPS), and RPMI 1640 were purchased from Sigma (St Louis, MO, USA). A De Man, Rogosa, and Sharpe (MRS) medium was purchased from BD (Franklin Lakes, NJ, USA). An antibody for NF-κB was purchased from Cell Signaling Technology (Danvers, MA, USA). Antibodies for CD11c, Iba1, cAMP-response element binding protein (CREB), p-CREB, and BDNF were purchased from Abcam (Cambridge, UK). Antibodies for IL-IR and NeuN were purchased from Millipore (Burlington, MA, USA). Alexa Fluor 488 was purchased from Invitrogen (Carbsband, CA, USA). Enzyme-linked immunosorbent assay (ELISA) kits for IL-1β, IL-6, and corticosterone were purchased from Ebioscience (Atlanta, GA, USA). A QIAamp Fast DNA stool mini kit was purchased from Qiagen (Hilden, Germany). EasyTaq DNA polymerase and 100 bp plus II DNA ladder was purchased from TransGen Biotech (Beijing, China). TB Green^®^ Premix Ex Taq™ II was purchased from Takara Bio (Shiga, Japan).

### 2.2. Selection of IL-1β Expression-Suppressing Lactobacillus gasseri NK109 and Its Dosage Regimen

The IL-1β expression-suppressing NK109, which did not exhibit hemolytic activity in 7% *v*/*v* sheep blood-contained blood agar plates, were screened from the human fecal bacteria strain collection in LPS-stimulated macrophages, as previously reported [14]. Bacteria, including NK109, were cultured in commercial media for probiotics, including MRS broth, centrifuged (5000× *g*, 4 °C, 20 min), and washed with saline, as previously reported [15]. Tyndallized NK109 (tNK109) was prepared by heating 90 °C for 30 min twice. The collected cells were used for further experiments.

To measure the dosage of NK109 in an animal experiment, NK109 at doses of 1 × 10^8^ and 1 × 10^9^ colony-forming units (CFUs)/mouse/day was orally gavaged once a day for 5 days in the mice with *Escherichia coli* K1 (1 × 10^9^ CFUs/mouse/day)-induced depression and cognitive impairment. Depressive and cognitive behaviors were estimated in a tail suspension test (TST) and a Y-maze task, respectively, as the previously reported [5,6]. NK109 at a dosage of 1 × 10^9^ CFUs/mouse/day significantly alleviated depression, cognitive impairment, and BDNF and IL-1β expression more strongly than one at a dosage of 1 × 10^8^ CFUs/mouse/day (Appendix A).

### 2.3. Culture of Macrophage Cells

The mice were injected intraperitoneally with sodium thioglycolate. Macrophage cells were collected from the peritoneal cavity 4 days after injection and suspended in RPMI 1640 containing 10% fetal bovine serum and 1% antibiotics (RFA). Suspended cells were seeded on a 12-well plate, incubated at 37 °C for 12 h, and washed with RFA, as reported previously [15]. To measure the expression of IL-1β and the activation of NF-κB, macrophages were incubated with LPS (80 ng/mL) in the absence or presence of isolated gut bacteria (1 × 10^5^ CFUs/mL) for 90 min (for p-p65 and p65) or 20 h (for IL-1β) [15]. The IL-1β level was assayed by ELISA and p-p65 and p65 levels were assayed by immunoblotting.

### 2.4. Animals

Germ-free C57BL/6J male mice (18–20 g, 5 weeks old) were purchased from Clea Japan Inc. (Tokyo, Japan) and bred under aseptic conditions according to The Guidelines for Laboratory Germ-free Animals Care and Usage. SPF C57BL/6 male mice (19–21 g, 5 weeks old) were purchased from Koatech (Pyungtaek-shi, Korea). All mice were maintained in plastic cages with the raised wire floor at 20–22 °C and 50 ± 10% humidity, and fed standard laboratory chow and water ad libitum. The mice were acclimatized for one week before usage in the experiments. All animal experiments were approved by The Committee for the Care and Use of Laboratory Animals in Kyung Hee University (Seoul, Korea) and carried out according to The Kyung Hee University Guidelines for Laboratory Animals Care and Usage (IACUC No., KHSASP-18-115 and 19-290).

### 2.5. Cognitive Impairment- and Depression-Induced Mice

The mice with cognitive impairment and depression were prepared, as reported previously [5,6]. First, to examine the effect of NK109 on the generation of neuropsychiatric disorder in germ-free mice, the mice were separated randomly into vehicle-treated (CoV), *Escherichia coli* K1-treated (EcV), and NK109-treated groups (ENK) in K1-treated mice. Each group consisted of four mice. The EcV and ENK groups were gavaged orally with the *Escherichia coli* K1 (1 × 10^7^ CFUs, suspended in 100 μL of saline) daily for 5 days. The CoV group was treated with the vehicle (saline) instead of the K1 suspension. From the next day, the ENK group was orally administered NK109 (1 × 10^9^ CFUs/mouse/day) daily for 5 days. The CoV and EcV groups were orally treated with saline instead of the test agents.

Second, to examine the effect of NK109 on the cognitive impairment and depression in the SPF mice, the mice were randomly separated into CoV, EcV, and ENK groups. Each group consisted of six mice. The EcV and ENK groups were orally treated with the K1 suspension (1 × 10^9^ CFUs, in 100 μL of saline) daily for 5 days. The CoV group was administered the vehicle instead of the K1 suspension. From the next day, the ENK group was orally administered NK109 (1 × 10^9^ CFUs/mouse/day) daily for 5 days. The CoV and EcV groups were orally gavaged with the vehicle.

Third, to understand the effect of NK109 on the vagus nerve-mediated gut–brain signaling, we prepared the SPF mice with celiac vagotomy, as reported previously [20]. The mice were separated into vehicle-treated group in sham mice (CoV), *Escherichia coli* K1-treated (vEcV) and NK109-treated groups in K1-treated mice with vagotomy (vENK) vehicle-treated group in sham mice (CoV), Escherichia coli K1-treated (vEcV) and NK109-treated groups in K1-treated mice with vagotomy (vENK). Each group consisted of six mice. The vCoV, vEc, and vENK groups were vagotomized. The vEcV and vENK groups were orally gavaged with K1 (1 × 10^9^ CFUs, suspended in 100 μL of saline) daily for 5 days. The vCoV group was administered with the vehicle instead of the K1 suspension. The vENK group was orally administered NK109 (1 × 10^9^ CFUs/mouse/day) daily for 5 day from 24 h after the final gavage of K1. The vEcV group was orally administered the vehicle instead of NK109.

Fourth, to examine the effects of tNK109 and its lysate on cognitive impairment and depression, NK109 was lysed by ultrasonication, centrifuged (5000× *g*, 20 min, 4 °C), and its supernatant (NC) and precipitate (NM) fractions were prepared. The SPF mice were randomly divided into the vehicle-treated (CoV), *Escherichia coli* K1-treated (EcV), tyndallized NK109-treated (EtN), NK109 sonicate supernatant fraction-treated (ENC), and NK109 sonicate precipitate fraction-treated (ENM) groups in K1-treated mice vehicle-treated (CoV), Escherichia coli K1-treated (EcV), tyndallized NK109-treated (EtN), NK109 sonicate supernatant fraction-treated (ENC), and NK109 sonicate precipitate fraction-treated (ENM) groups in K1-treated mice. Each group consisted of six mice. The EcV, EtN, ENC, and ENM groups were orally gavaged with the K1 suspension (1 × 10^9^ CFUs, suspended in 100 μL of saline) once a day for 5 days. The CoV group was treated with saline instead of K1 for 5 days. From the next day, the EtN, ENC, and ENM groups were orally administered tNK109 (1 × 10^9^ CFUs/mouse/day), NC (supernatant fraction of NK (1 × 10^9^ CFUs) lysate/mouse/day), and NM (precipitate fraction of NK (1 × 10^9^ CFUs) lysate/mouse/day) for 5 days, respectively. The mice of the CoV and EcV groups were orally gavaged with the vehicle (saline).

Depressive and cognitive behaviors were carried out 24 h after the final administration of the test agents. The mice were then killed 18 h after the final behavioral tasks. The blood, colon, and hippocampus were removed and stored at −80 °C until used in experiments.

### 2.6. Behavioral Tasks

For the assay of cognitive behaviors, the Y-maze task was carried out in a three-arm horizontal maze, which was 3 cm wide and 40 cm long with 12 cm-high walls [6,21]. A mouse was initially placed within one arm. The sequence and number of arm entries were then monitored for 8 min. A spontaneous alternation was indicated as entries into all three arms on consecutive choices and calculated as the ratio (%) of spontaneous to possible alternations.

For the assay of depression-like behaviors, the forced swimming test (FST) and TST were carried out, as reported previously [6]. The TST was carried out on the edge of a table above 30 cm for 5 min. When the mice did not move and hanged passively, they were declared to be immobile. The FST was carried out in a round transparent plastic jar (20 × 40 cm^3^) filled with fresh water (25 °C) to a height of 25 cm for 5 min [6]. When the mice remained floating in the water without struggling, they were declared to be immobile.

### 2.7. Myeloperoxidase Activity Assay, Immunoblotting, and ELISA

Myeloperoxidase activity was measured in colon homogenate supernatant, as reported previously [14]. BDNF, CREB, p-CREB, p65, p-p65, and β-action expression levels were determined in the supernatants of the tissue homogenates by immunoblotting, as reported previously [16]. Corticosterone, IL-6, and IL-1β levels were determined in the supernatant of the colon and hippocampus homogenate and plasma using commercial ELISA kits [16].

### 2.8. Immunohistochemistry

For the immunohistochemistry analysis, the mice were perfused transcardially with 4% paraformaldehyde. Hippocampi and colons were removed, post-fixed with 4% paraformaldehyde, immersed in 30% sucrose solution, frozen, and sectioned using a cryostat, as reported previously [20]. The sections were incubated with antibodies for NF-κB (1:100), CD11c (1:200), Iba1 (1:200), BDNF (1:200), IL-IR (1:200), and/or NeuN (1:200) and incubated with the Alexa Fluor 488 (1:200)- or Alexa Fluor 594 (1:200)-conjugated secondary antibody [6]. Cell nuclei were stained with DAPI. The immunostained section was observed with a confocal microscope.

### 2.9. Microbiota Sequencing

Genomic DNA of mouse fecal bacteria was extracted from fresh mouse stools using a QIAamp DNA stool mini kit, as reported previously [6,19]. Genomic DNA amplification was carried out using barcoded primers, which targeted the V4 region of the bacterial 16S *rRNA* gene. Each amplicon sequencing was carried out by using Illumina iSeq 100 (San Diego, CA, USA). The pyrosequencing reads were deposited in the NCBI’s short read archive under accession number PRJNA622393.

### 2.10. Statistical Analysis

Experimental data are indicated as mean ± standard deviation (SD) and analyzed by Graph-Pad Prism 8 (GraphPad Software Inc., San Diego, CA, USA). The significance was analyzed using the unpaired *t*–test (*p* < 0.05). All *p*-values for the experimental data in the present study are indicated in Appendix A.

## 3. Results and Discussion

To examine whether IL-1β expression-suppressing probiotics could alleviate cognition impairment and depression, we selected IL-1β expression-inhibiting probiotics from a human fecal bacterial strain collection. Of the tested probiotics, NK109 strongly suppressed K1-induced IL-1β expression and NF-κB activation in macrophage cells (Figure 1A,B). NK109 also suppressed IL-1β expression and NF-κB activation in LPS-stimulated macrophage cells (Appendix A). NK109 was identified as *Lactobacillus gasseri* on the basis of the results of gram staining, 16S *rDNA* sequencing, and API 50 CHL kit.

Next, we examined the effect of NK109 at a dose of 1 × 10^7^ CFUs/mouse/day on K1-induced depression in the germ-free mice (Figure 2A). Exposure to K1 significantly increased the depression-like behaviors in the TST to 257.1% of the control group. Oral gavage of NK109 alleviated K1-induced depressive behaviors to 141.1% of the control group. Treatment with NK109 also decreased K1-induced IL-1β expression, NF-κB^+^/Iba1^+^ (activated microglia) and IL-1R^+^ cell populations, and NF-κB activation in the hippocampus, while the K1-suppressed BDNF^+^/NeuN^+^ cell population and BDNF expression increased (Figure 2B–F). Furthermore, NK109 treatment decreased K1-induced myeloperoxidase activity, the IL-1β expression, the NF-κB^+^/CD11c^+^ cell population, and NF-κB activation in the colon (Figure 2G–J).

To prepare the mice with psychiatric disorders, we orally gavaged K1 at dosages of 1 × 10^7^, 1 × 10^8^, and 1 × 10^9^ CFUs/mouse/day in the SPF mice and measured the depression-like behavior in the TST (data not shown), as previously reported [6]. Oral gavage of K1 at a dose of 1 × 10^9^ CFUs/mouse/day most strongly caused depression. Therefore, we investigated the effect of NK109 on the occurrence of cognitive impairment and depression in the SPF mice with K1 (orally gavaged at a dose of 1 × 10^9^ CFUs/mouse/day)-induced psychiatric disorders (Figure 3). Exposure to K1 caused cognitive impairment in the Y-maze task to 82.6% of the control group (Figure 3A). However, oral gavage of NK109 (1 × 10^9^ CFUs/mouse/day) alleviated K1-induced cognitive impairment in the Y-maze task to 109.6% of the control group. Exposure to K1 also significantly increased depressive behaviors in the TST and FST to 139.4% and 298.4% of the control group, respectively (Figure 3B,C). Oral administration of NK109 reduced the K1-increased immobility time in the TST and FST to 81.0% and 148.9% of the control group, respectively. However, oral administration of NK109 did not affect the cognitive function in the control mice (Appendix A). K1 treatment increased the activation of NF-κB and the expression of IL-1β and IL-6, as well as reduced the expression of BDNF in the hippocampus (Figure 3D,E). However, oral gavage of NK109 decreased the K1-induced NF-κB activation, IL-1β and IL-6 levels, and NF-κB^+^/Iba1^+^ and IL-1R^+^ cell populations, as well as increased the K1-suppressed BDNF^+^/NeuN^+^ cell population, CREB phosphorylation, and BDNF expression in the hippocampus (Figure 3D–I). Furthermore, oral gavage of NK109 reduced the K1-induced IL-6 and corticosterone levels in the blood (Figure 3J,K). Furthermore, to understand the vagus nerve-mediated effect of NK109 on K1-induced cognitive impairment and depression, we examined its effects in the celiac vagotomy-treated mice. Vagotomy itself did not cause significant memory impairment or depression in the Y-maze task or TST, respectively (Appendix A). K1 treatment caused cognitive impairment- and depression-like behaviors. Oral administration of NK109 weakly, but not significantly, alleviated K1-induced cognitive impairment in the Y-maze task and depression in the TST (Figure 3L,M). NK109 also increased hippocampal CREB phosphorylation and BDNF expression (Figure 3N). The ameliorating activity of NK109 against cognitive impairment and depression was lower in the vagotomy-operated mice than in the vagotomy-untreated mice.

Exposure to K1 caused colon shortening and increased myeloperoxidase activity, the NF-κB+/CD11c+ cell population, NF-κB activation, and IL-1β and IL-6 expression, resulting in colitis (Figure 4A–G). However, oral gavage of NK109 significantly suppressed K1-induced colitis: It reduced colon shortening, myeloperoxidase activity, NF-κB activation, the NF-κB+/CD11c+ cell population, and IL-1β and IL-6 expression.

Next, we investigated the effect of NK109 on the K1-induced gut microbiota alteration in the mice with K1-induced cognitive impairment and depression (Figure 3H–J, Appendix A). Oral gavage of K1 significantly increased the populations of Proteobacteria and Cyanobacteria, while the population of Bacteroidetes decreased. K1 treatment did not affect α-diversity, including estimated operational taxonomic unit (OTU) richness and Shannon’s diversity index, compared to those in the control mice, while K1 treatment significantly shifted the β-diversity, as shown by the principal coordinate analysis (PCoA) based on Jansen–Shannon. Oral administration of NK109 decreased the K1-induced β-diversity change, while the α-diversity was not affected. NK109 treatment decreased the K1-induced populations of Proteobacteria and Cyanobacteria and increased the K1-suppressed population of Bacteroidetes. Oral administration of NK109 increased the K1-suppressed populations of Bacteroidaceae and Muribaculaceae in the family level and Bacteroides, Muribaculaceae_uc, and PAC001063_g at the genus level and *Bacteroides coccae* and PAC00163_g_uc at the species level. However, NK109 treatment decreased the K1-induced population of Christensenellaceae, Desulfovibrionaceae, and Rhodospirillaceae in the family and Alloprevotella, FR888536_g, Helicobacter, and LARJ_g at the genus level and LARJ_g, FJ880724_s, and the *Helicobacter rodentium* group at the species level.

Next, to evaluate the biological difference between the live and tyndallized probiotics, we tyndallized NK109, lysed by ultrasonication, centrifuged, and examined the effects of tNK109 and its supernatant (NC) and precipitate fractions (NM) on K1-induced cognitive impairment and depression in the SPF mice (Figure 5). Their treatment significantly alleviated K1-suppressed spontaneous alteration in the Y-maze task and the K1-increased immobility time in the TST and FST (Figure 5A–C). Of the tested tNK109, NC, and NM, tNK109 and NM more strongly alleviated K1-induced cognitive impairment and depression than did NC. They also suppressed K1-induced hippocampal IL-1β and IL-6 expression, NF-κB^+^/Iba1^+^ and IL-1R^+^ cell populations, and NF-κB activation, as well as increased the K1-suppressed hippocampal BDNF^+^/NeuN^+^ cell population and BDNF expression (Figure 5D–I). K1-induced IL-1β and corticosterone levels decreased in the blood by oral administration of tNK109, NC, or NM (Figure 5J,K). Treatment with tNK109 or NM also suppressed the K1-induced colitis markers: They reduced colon shortening and NF-κB activation, as well as decreased myeloperoxidase activity, IL-1β, IL-6, and TNF-α levels, and the NF-κB^+^/CD11c^+^ cell population (Figure 6A–G).

The prevalence of psychiatric disorders is significantly higher in patients with inflammatory bowel disease (IBD) than in healthy people [22]. Exposure to stresses, including immobilization, causes psychiatric disorders with colitis and gut dysbiosis via the HPA axis activation [3,4]. Gut dysbiosis and inflammation induce the occurrence of psychiatric disorders through activation of the MGB axis [5,21,22]. Exposure of SPF mice to stressors causes gut dysbiosis: They increase the Proteobacteria population in the gut microbiota [5,16]. A gut commensal *Lactobacillus johnsonii* alleviates anxiety-like behaviors with gut dysbiosis in mice [5,8]. The brain bidirectionally communicates to the gut microbiota through the HPA and MGB axes [1,2].

In the present study, exposure to *Escherichia coli* K1 significantly caused psychiatric disorders, including depression and memory impairment, and IBD-like colitis in the SPF or germ-free mice, as reported previously [5,6]. Exposure to *Escherichia coli* (at a dose of 1 × 10^7^ CFUs/day/mouse) caused depression more strongly in the germ-free mice than in the SPF mice. The occurrence of depression-like behaviors by exposure of the germ-free mice to *Escherichia coli* at a dose of 1 × 10^7^ CFUs/day/mouse was similar to that in the SPF mice exposed to *Escherichia coli* at a dose of 1 × 10^9^ CFUs/day/mouse. This suggests that normal gut microbiota may defend the colonization of pathogenic microbes such as *Escherichia coli* in the gastrointestinal tract. We also found that K1 caused neuroinflammation and colitis: It increased the expression of IL-1β and IL-1 receptor (IL-1R) in the hippocampus and IL-1β expression in the colon. The overgrowth of gut commensal *Escherichia coli* by immobilization stress, a colitis inducer, or oral gavage of *Escherichia coli* causes colitis and neuroinflammation: *Escherichia coli* induces the expression of IL-1β and IL-6 in the hippocampus and colon [5]. Stress significantly induces the expression of IL-1β in the hippocampus and the administration of IL-1R antagonist alleviates depression-like behaviors [7]. An anti-inflammatory therapy in patients with IBD alleviates cognitive function [23]. These findings suggest that K1 may cause psychiatric disorders by enhancing the expression of pro-inflammatory cytokines, particularly IL-1β, and their receptors, such as IL-1R, in the hippocampus, and the suppression of IL-1β expression can alleviate psychiatric disorders. 

We also found that oral gavage of NK109, which significantly inhibited IL-1β expression in K1-stimulated macrophages, alleviated K1-induced cognition impairment and depression in the mice. Furthermore, NK109 suppressed hippocampal inflammation in the K1-treated mice: It decreased NF-κB activation and IL-1β expression in the hippocampus, while the IL-1R^+^ and NF-κB^+^/Iba1^+^ cell (activated microglia) populations increased. NK109 alleviated K1-induced colitis: It suppressed the expression of IL-1β and activation of NF-κB in the colon. These results suggest that NK109 can mitigate neuroinflammation and colitis by inhibiting Il-1β expression through the regulation of the NF-κB signaling pathway, resulting in the amelioration of cognition impairment and depression.

NK109 treatment increased K1-suppressed BDNF expression and the BDNF+/NeuN+ cell population in the hippocampus. NK109 also suppressed K1-induced expression of IL-6, which may be associated in the occurrence of neuropsychiatric disorders [24], and corticosterone in the blood. Neuroinflammation suppresses BDNF expression in the brain of mice [25]. Hippocampal BDNF expression is associated with the outbreak of cognitive impairment and depression [25,26,27]. These findings suggest that NK109 may alleviate cognitive impairment and depression by regulating NF-κB-mediated BDNF expression.

Moreover, the ameliorating effect of NK109 against K1-induced cognitive impairment and depression was partially attenuated by celiac vagotomy. Vagotomy suppresses the ameliorating effect of *Lactobacillus rhamnosus* against anxiety and depression [28]. These findings suggest that NK109 may alleviate neuropsychiatric disorders by modulating the immune response and vagus nerve-mediated gut–brain signaling through the regulation of the pro-inflammatory cytokine expression and gut bacteria byproducts.

NK109 attenuated K1-induced gut microbiota alteration: It suppressed the Proteobacteria population. Excessive growth of Proteobacteria, particularly Enterobacteriaceae, causes colitis, resulting in the occurrence of neuroinflammation. NK109 alleviated K1-induced colitis and neuroinflammation and colitis with gut dysbiosis in the mice, leading to the attenuation of cognitive impairment and depression. *Bifidobacterium breve* strain A1 improves Aβ protein-induced cognition impairment and neuroinflammation in mice [29]. *Lactobacillus plantarum* C29 alleviates the cognition impairment in aged mice and patients with cognitive decline by inducing BDNF expression [27,28]. *Lactobacillus mucosae* NK41 significantly mitigates *Escherichia coli*-induced neuropsychiatric disorders, neuroinflammation, and colitis in rodents [6]. These findings suggest that NK109 can alleviate the generation of cognitive impairment and depression by suppressing neuroinflammation and colitis through the modulation of gut microbiota.

tNK109 and its lysates NC and NM alleviated K1-induced depression and cognitive impairment with colitis in the mice by regulating NF-κB-mediated BDNF and IL-1β expression. Their effects were comparable to those of live NK109, suggesting that NK109, even if killed and lysed, can regulate NF-κB-mediated BDNF and IL-1β expression by the modulation of gut dysbiosis, resulting in the amelioration of neuropsychiatric disorders.

## 4. Conclusions

NK109 significantly alleviated *Escherichia coli* K1-induced cognitive impairment- and depression-like behaviors in germ-free and SPF mice by regulating the immune response through NF-κB-involved BDNF expression, IL-1β expression, and vagus nerve-mediated gut–brain signaling. NK109 treatment also mitigated *Escherichia coli*-induced colitis and gut dysbiosis. tNK109 and its lysates NC and NM also alleviated cognitive impairment- and depression-like behaviors and colitis. Finally, the attenuation of gut dysbiosis and IL-1β expression by beneficial bacteria, including NK109, can alleviate neuropsychiatric disorders such as cognitive impairment and depression.

## Figures and Tables

**Figure 1 nutrients-12-03441-f001:**
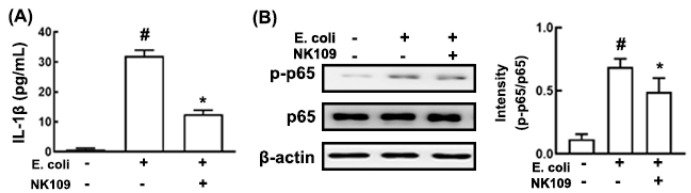
Effect of *Lactobacillus gasseri* NK109 on the *Escherichia coli* K1-induced interleukin (IL)-1β and nuclear factor (NF)-κB activation in macrophage cells. Effect on IL-1β expression (**A**) and NF-κB activation (**B**) on macrophage cells. Macrophage cells (1 × 10^6^/mL) isolated from the peritoneal cavity were incubated with NK109 (1 × 10^5^ colony-forming units (CFUs)/mL) in the absence or presence of *Escherichia coli* K1 (1 × 10^5^ CFUs/mL). Data values are described as mean ± standard deviation (SD) (*n* = 4). ^#^
*p* < 0.05 vs. group not treated with *E. coli* alone. * *p* < 0.05 vs. group treated with *E. coli* alone.

**Figure 2 nutrients-12-03441-f002:**
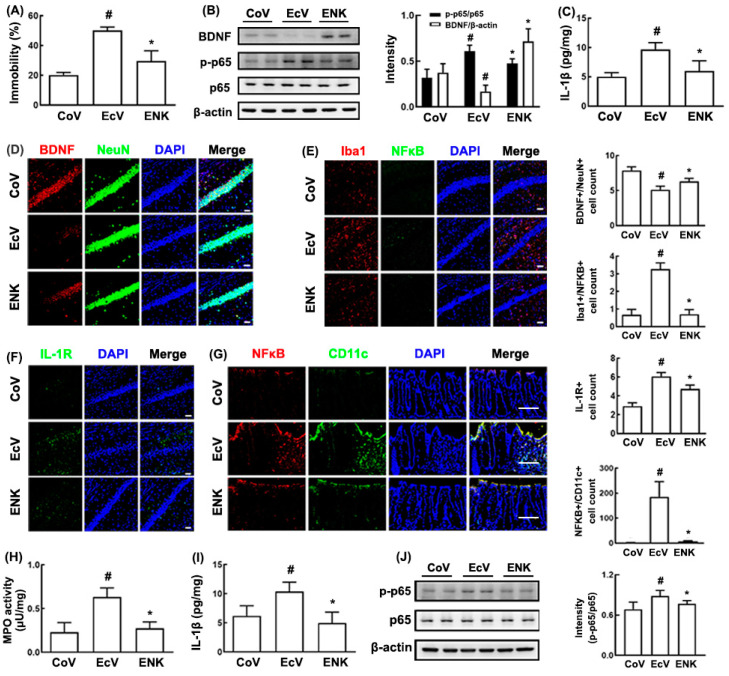
Effect of NK109 on the *Escherichia coli* K1-induced cognitive impairment and colitis in the germ-free mice. (**A**) Effect on the immobility time in the tail suspension test (TST). (**B**) Effect on the brain-derived neurotropic factor (BDNF) expression and NF-κB activation in the hippocampus. (**C**) Effect on the IL-1β level in the hippocampus. Effects on the BDNF^+^/neuronal nuclei (NeuN)^+^ (**D**), NF-κB^+^/Iba1^+^ (**E**), and IL-1R^+^ cell (**F**) populations in the hippocampus. Effect on the myeloperoxidase (MPO) activity (**G**), IL-1β expression (**H**), the NF-κB^+^/CD11c^+^ cell population (**I**), and NF-κB activation (**J**) in the colon. *Escherichia coli* K1-treated (EcV) and K1/NK109-treated (ENK) groups were exposed to K1 (1 × 10^9^ CFUs/mouse/day) daily for 5 days and thereafter treated with NK109 (for ENK, 1 × 10^9^ CFUs/mouse/day) or the vehicle (for EcV) for 5 days. The control group (CoV) was treated with saline instead of K1 and NK109. Data values were are as mean ± SD (*n* = 4). ^#^
*p* < 0.05 vs. CoV. * *p* < 0.05 vs. EcV. Dapi, 4′, 6-diamidino-2-phenylindole dihydrochloride; Iba1, ionized calcium-binding adapter molecule 1; NF-κB, nuclear factor-κB; IL-1R, interleukin 1 receptor; IL-1β, interleukin 1β; CD11c, a type I transmembrane protein.

**Figure 3 nutrients-12-03441-f003:**
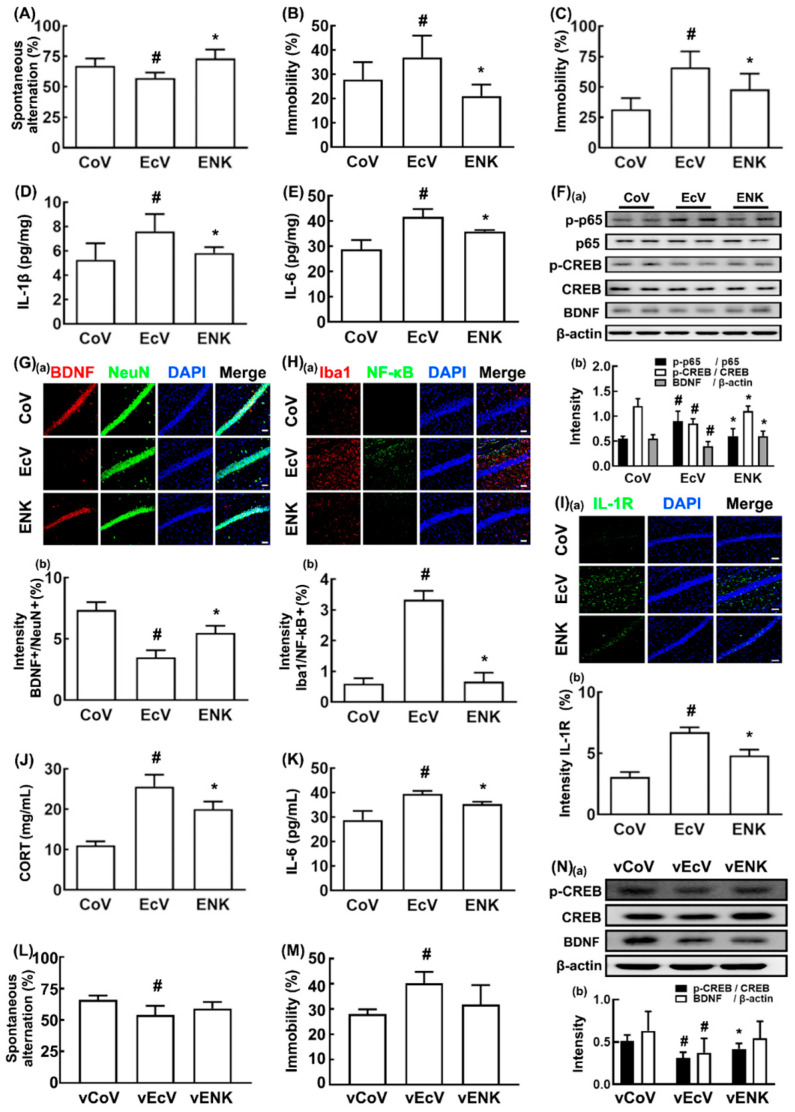
NK109 alleviated K1-induced cognitive impairment and depression in the mice. (**A**) Effect on cognitive impairment in the Y-maze task. Effect on depressive behaviors in the TST (**B**) and forced swimming test (FST) (**C**). Effect on the hippocampal IL-1β (**D**) and IL-6 levels (**E**,**F**) Effect on the hippocampal BDNF expression and NF-κB activation (**a**) and their intensities (**b**). Effect on the hippocampal BDNF^+^/NeuN^+^ ((**G**): (**a**) Confocal microscope photo; (**b**) their intensities), NF-κB^+^/Iba1^+^ ((**H**): (**a**) Confocal microscope photo; (**b**) their intensities), and IL-1R^+^ cell ((**I**): (**a**) Confocal microscope photo; (**b**) their intensities) populations. Effect on the blood corticosterone (**J**) and IL-6 levels (**K**), assessed by ELISA. Effect on cognitive impairment in the Y-maze task (**L**) and depressive behaviors in the TST (**M**) and hippocampal BDNF expression and cAMP-response-element-binding protein (CREB) phosphorylation ((**N**): (**a**) Immunoblotting photo; (**b**) their intensities). The vehicle-treated group in *Escherichia coli* K1-treated mice (EcV), vehicle-treated group in *Escherichia coli* K1-treated mice with vagotomy (vEcV), NK109-treated group in *Escherichia coli*-treated mice (ENK), NK109-treated in *Escherichia coli* K1-treated mice with vagotomy (vENK) were orally gavaged with the K1 (1 × 10^9^ CFUs/mouse/day). The CoV and vCoV groups were treated with the vehicle instead of the K1 suspension. From the next day, the ENK and vENK groups were orally gavaged with NK109 (1 × 10^9^ CFUs/mouse/day). The vehicle-treated in mice (CoV), vehicle-treated in sham mice (vCoV), vehicle-treated in *Escherichia coli* K1-treated mice (EcK), vehicle-treated in *Escherichia coli* K1-treated mice with vagotomy (vEcK) were orally gavaged with the vehicle. (**A**–**K**) Experiments for the CoV, EcV, and ENK groups were performed in the mice without vagotomy. Experiments for the vCoV, vEcV, and vENK groups were performed in the mice with vagotomy. Data values are described as mean ± SD (*n* = 6). Means with same letters are not significantly different (*p* < 0.05). ^#^
*p* < 0.05 vs. CoV or vCoV. * *p* < 0.05 vs. EcV or vEcV. CORT, corticosterone.

**Figure 4 nutrients-12-03441-f004:**
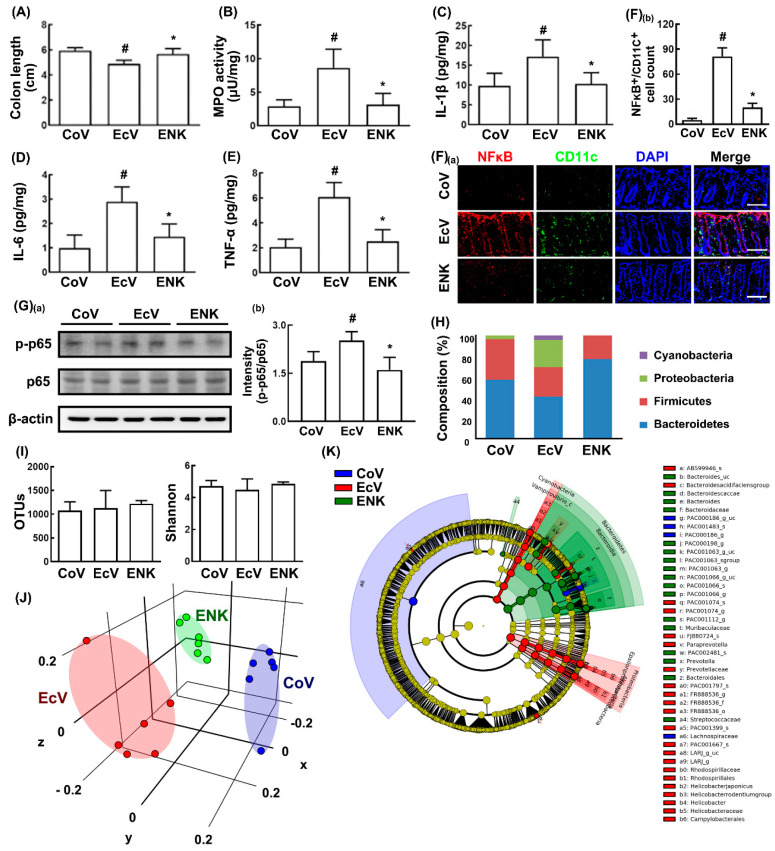
NK109 alleviated K1-induced colitis and gut microbiota alteration in the mice. Effects on the colon length (**A**), myeloperoxidase (MPO) activity (**B**), IL-1β (**C**), IL-6 (**D**), and TNF-α (**E**) expression. (**F**) Effect on the colon NF-κB^+^/CD11c^+^ cell population (**a**) and its intensity (**b**). (**G**) Effect on the NF-κB activation (**a**) and its intensity (**b**). Effects on the composition of gut microbiota: Phylum (**H**), operational taxonomic units (OTUs) and Shannon (**I**), PCoA plot based on based on Jansen–Shannon (**J**), and Cladogram generated by LEfSE showing significant differences in the abundance of gut microbial composition among the CoV (blue), EcV (red), and ENK (green) groups (**K**). The EcV and ENK groups were exposed to K1 (1 × 10^9^ CFUs/mouse/day) and thereafter treated with NK109 (for ENK, 1 × 10^9^ CFUs/mouse/day) or the vehicle (for EcV) for 5 days. The control group (CoV) was treated with the vehicle instead of K1 and NK109. Data values are described as mean ± SD (*n* = 6). ^#^
*p* < 0.05 vs. CoV. * *p* < 0.05 vs. EcV.

**Figure 5 nutrients-12-03441-f005:**
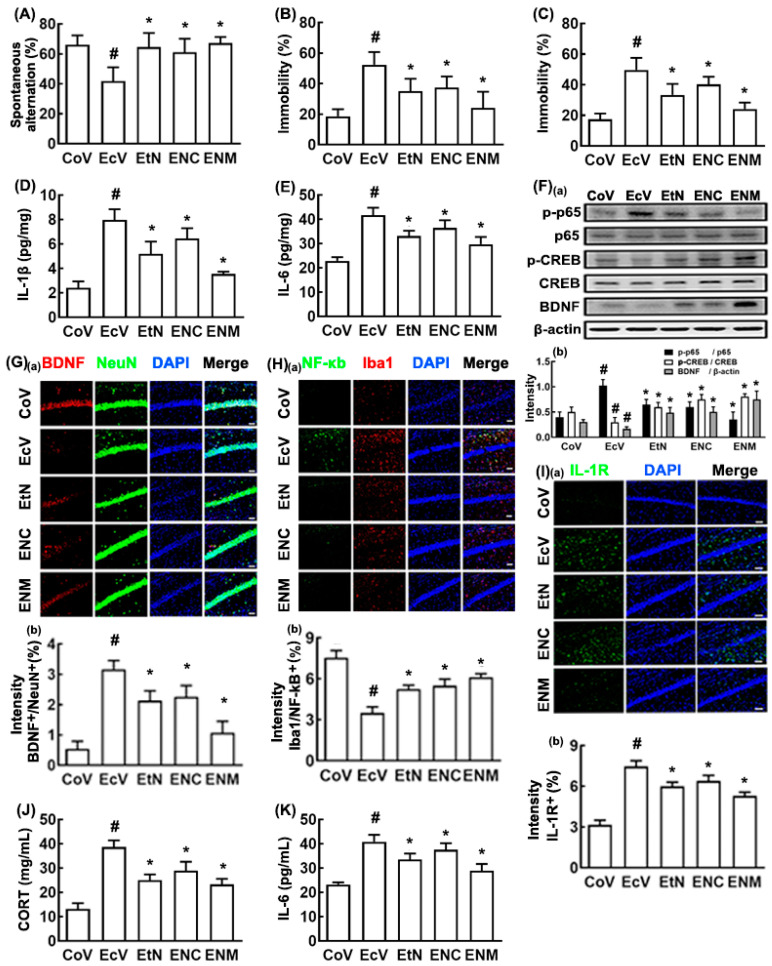
NK109 and its lysates NC and NM alleviated K1-induced cognitive impairment and depression in the mice. (**A**) Effects on the cognitive impairment in Y-maze task. Effects on the depressive behaviors in the TST (**B**) and FST (**C**). Effects on the hippocampal IL-1β (**D**) and IL-6 levels (**E**). (**F**) Effects on the hippocampal BDNF expression and NF-κB activation (**a**) and their intensities (**b**). Effect on the hippocampal BDNF^+^/NeuN^+^ ((**G**): (**a**) Confocal microscope; (**b**) their intensities), NF-κB^+^/Iba1^+^ ((**H**): (**a**) Confocal microscope photo; (**b**) their intensities), and IL-1R^+^ cell ((**I**): (**a**) Confocal microscope; (**b**) their intensities) populations. Effects on the blood corticosterone (**J**) and IL-6 levels (**K**), assessed by ELISA. *Escherichia coli* K1-treated (EcV), tyndallized NK109-treated (EtN), NK109 sonicate supernatant fraction-treated (ENC) groups were exposed to K1 and thereafter treated with test agents (EcV, vehicle; EtN, tNK109; ENC, supernatant fraction (NC) of NK109 (1 × 10^9^ CFU) lysate/mouse/day; ENM, precipitate fraction (NM) of NK109 (1 × 10^9^ CFU) lysate/mouse/day) daily for 5 days. Control group (CoV) was treated with saline instead of K1 and NK109. Data values were described as mean ± SD (*n* = 6). ^#^
*p* < 0.05 vs. CoV. * *p* < 0.05 vs. EcV.

**Figure 6 nutrients-12-03441-f006:**
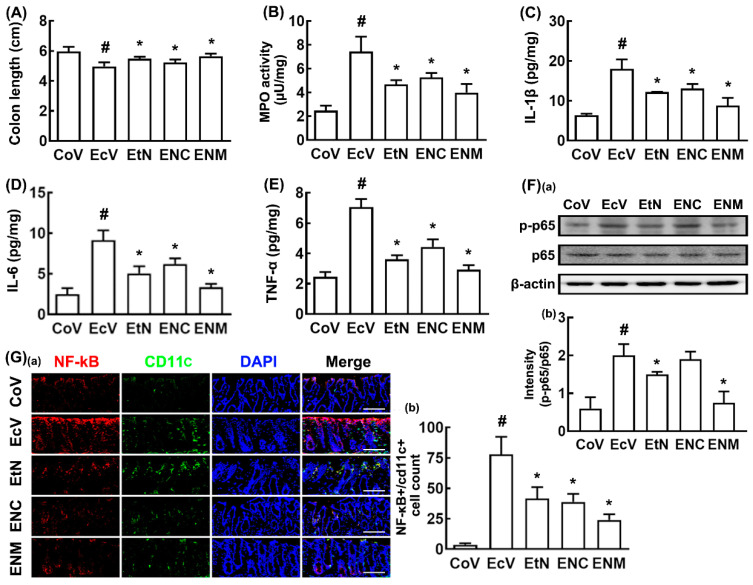
NK109 and its lysates alleviated K1-induced colitis in the mice. Effects on the colon length (**A**), myeloperoxidase (MPO) activity (**B**), IL-1β (**C**), IL-6 (**D**), TNF-α expression (**E**), NF-κB activation ((**F**): (**a**) Immunoblotting photo; (**b**) their intensities), and the NF-κB^+^/CD11c^+^ cell population ((**G**): (**a**) Confocal microscope photo; (**b**), their intensities) in the colon. The mice of the EcV, EtN, ENC, and ENM groups were exposed to K1 (1 × 10^9^ CFUs/mouse/day) daily for 5 days and thereafter treated with the test agents (EcV, vehicle; EtN, 1 × 10^9^ CFUs/mouse/day of tNK109; ENC, supernatant fraction (NC) of NK109 (1 × 10^9^ CFUs) lysate/mouse/day; ENM, precipitate fraction (NM) of NK109 (1 × 10^9^ CFUs) lysate/mouse/day) daily for 5 days. The control group (CoV) was treated with the vehicle instead of K1 and NK109. Data values are described as mean ± SD (*n* = 6). ^#^
*p* < 0.05 vs. CoV. * *p* < 0.05 vs. EcV.

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
