# Peer review of "A Probiotic Lactobacillus gasseri Alleviates Escherichia coli-Induced Cognitive Impairment and Depression in Mice by Regulating IL-1β Expression and Gut Microbiota"

_nutrients, 2020, doi:10.3390/nu12113441_

Round 1
Reviewer 1 Report
In this manuscript, the authors concluded that NK109 alleviated neuropsychiatric disorders and colitis by modulating IL-1b expression, gut microbiota, and vagus nerve-mediated gut-brain signaling.
However, I found that authors can not draw a certain conclusion on the basis of the measurement of a few markers. A behavior test was needed to confirm those results. Moreover, the quality of the WB is not optimal and the original scans have to be reported. Same comment for the graphs.
Please see below some other comments:
Lines 41-43: Although lactobacilli are considered as beneficial bacteria, there are several studies describing a significant increase of Lactobacillus spp., in the context of obesity. I will pay attention when talking about their ability to suppress body weight gain.
Lines 59-65: There are different spelling mistakes and missing parts Please check it.
Line 72: 90oC has to be replaced with 90°C. In vitro and in vivo test goes in Italique.
Line 170: Statistical analysis
Why authors decided to express the results as mean ± standard deviation, rather than SEM?
Line 182: Typos error, 107 has to be replaced with 107. Based on what authors decided to use this concentration?
Figure 1: Please check the graphs, not the same size, and in line with the other graphs.
Are the authors sure about the statistically significant difference in the intensity (p-p65/p65)? On the WB analysis, I cannot clearly see this difference. Moreover, authors should report the original scan of the WB.
Line 183: Exposure to K1 significantly increased the depression-like 184 behaviors in the TST to 257.1% of the control group.
You cannot clearly see this in the graph. I do not see 257.1%.
Author Response
In this manuscript, the authors concluded that NK109 alleviated neuropsychiatric disorders and colitis by modulating IL-1b expression, gut microbiota, and vagus nerve-mediated gut-brain signaling.
However, I found that authors can not draw a certain conclusion on the basis of the measurement of a few markers. A behavior test was needed to confirm those results. Moreover, the quality of the WB is not optimal and the original scans have to be reported. Same comment for the graphs.
Please see below some other comments:
Lines 41-43: Although lactobacilli are considered as beneficial bacteria, there are several studies describing a significant increase of Lactobacillus spp., in the context of obesity. I will pay attention when talking about their ability to suppress body weight gain.
-->Thank you. We revised it according to your comment. We described the anti-Helicobacter pylori activity of Lactobacilli instead of it.
Lines 59-65: There are different spelling mistakes and missing parts Please check it.
-->Thank you. We revised them.
Line 72: 90oC has to be replaced with 90°C. In vitro and in vivo test goes in Italique.
-->Thank you. We revised them.
Line 170: Statistical analysis
Why authors decided to express the results as mean ± standard deviation, rather than SEM?
-->Thank you comment. When we described mean+SEM, SD values were lower than SEM. We can see the data variations. And we indicated all p-values for all experiment data in Supplementary material Table S1.
Line 182: Typos error, 107 has to be replaced with 107. Based on what authors decided to use this concentration?
-->Thank you. We revised them.
Figure 1: Please check the graphs, not the same size, and in line with the other graphs.
Are the authors sure about the statistically significant difference in the intensity (p-p65/p65)? On the WB analysis, I cannot clearly see this difference. Moreover, authors should report the original scan of the WB.
-->Thank you. We did it several times. We confirmed its results almost were similar. We added the recently scanned results.
Line 183: Exposure to K1 significantly increased the depression-like behaviors in the TST to 257.1% of the control group. You cannot clearly see this in the graph. I do not see 257.1%.
-->Thank you. We confirmed it according to your comment. It was correct.
Reviewer 2 Report
The point of the hypotheses about microbiota-gut-brain (MGB) axis, it was a very interesting paper that improved cognitive decline and depression by intervention with Lactobacillus Gasseri with caused by Escherichia coli. I would like to evaluate it as an integrated study that also uses the expression of each factor in the brain and intestinal tissues expression and behavioral experiments with germ-free animals and vagus nerve ligation experiments.
But there are several problems to be resolved points for revise.
Major point
- How about 6 animals in each group statistically? Please describe the statistical basis in a little more detail.
- In the whole figures、the legends last description
” Means with same letters are not significantly different (p < 0.05).”
I don't understand the meaning. Asterisk where there is a significant difference!
- L203: The data value is average ± SD (n = 4), but what about 6 animals each (in Fig 1 legends)?
- There is no description of materials and methods for measuring the length of the large intestine about in Fig. 3A.
Minor point
- L 61: “BD (Franklin Lakes, NJ). Antibodies for NF-κB was purchased from Cell Signaling (…). Antibodies 61 for CD11c, Iba1, and BDNF were purchased from Abcam (>>>>). Antibodies for IL-IR and NeuN were 62 purchased from Millipore (). Alexa Fluor 488 and 594 were purchased from Invitrogen (…. QIAamp “
Insufficient description in parentheses
- L143:What kind of test is TST? The first appearance should be full spelled out.
- The mark on the graph on the right side of Fig. 1D is difficult to understand. The usage guide of p-p65/p65 is black and BDNF/b-action is gray, but the graph is black and white.
- L245: “Bacteroidaceae and Muribaculaceae populations in the family level and Bacteroides,,,,,”The rest is not connected
- 2G)H), The relationship between figures and graphs is difficult to understand Isn’t it better to put together the photos of G) H) for all the graphs / NF-kB?
- L257: Fig2 legend “Experiments for CoV, EcV, and ENK groups were performed in mice without vagotomy. Experiments for vCoV, vEcV, and vENK groups were in mice with vagotomy. Data values were indicated as mean 258 ± SD (n = 6). Means with same letters are not significantly different (p < 0.05). “Is it about graph L)-M)? The author should describe in Fig 2 legends
- L401 induced colitis and gut dysbiosis. tNK109 and its lysates NC and NM also alleviated cognitive. What is tNK109? Should delete “t”, should not?
Author Response
Comments and Suggestions for Authors
The point of the hypotheses about microbiota-gut-brain (MGB) axis, it was a very interesting paper that improved cognitive decline and depression by intervention with Lactobacillus Gasseri with caused by Escherichia coli. I would like to evaluate it as an integrated study that also uses the expression of each factor in the brain and intestinal tissues expression and behavioral experiments with germ-free animals and vagus nerve ligation experiments.
But there are several problems to be resolved points for revise.
Major point
How about 6 animals in each group statistically? Please describe the statistical basis in a little more detail.
-->Thank you. We added all statistical p-value data in Table S1 of Supplementary information.
In the whole figures、the legends last description
” Means with same letters are not significantly different (p < 0.05).”
I don't understand the meaning. Asterisk where there is a significant difference!
-->Thank you. We revised them.
L203: The data value is average ± SD (n = 4), but what about 6 animals each (in Fig 1 legends)?
-->Thank you. We have the small facility for the experiment of germ-free animal revised them. Therefore, For GM mouse experiment, a group was 4 mice.
There is no description of materials and methods for measuring the length of the large intestine about in Fig. 3A.
-->Thank you. We revised them.
Minor point
L 61: “BD (Franklin Lakes, NJ). Antibodies for NF-κB was purchased from Cell Signaling (…). Antibodies 61 for CD11c, Iba1, and BDNF were purchased from Abcam (>>>>). Antibodies for IL-IR and NeuN were 62 purchased from Millipore (). Alexa Fluor 488 and 594 were purchased from Invitrogen (…. QIAamp “
-->Thank you. We revised them.
Insufficient description in parentheses
L143:What kind of test is TST? The first appearance should be full spelled out.
-->Thank you. We revised it in Line 81
The mark on the graph on the right side of Fig. 1D is difficult to understand. The usage guide of p-p65/p65 is black and BDNF/b-action is gray, but the graph is black and white.
-->Thank you. We revised them.
L245: “Bacteroidaceae and Muribaculaceae populations in the family level and Bacteroides,,,,,”The rest is not connected
-->Thank you. We revised it in Line 278.
2G)H), The relationship between figures and graphs is difficult to understand Isn’t it better to put together the photos of G) H) for all the graphs / NF-kB?
-->Thank you. We rearranged Figures.
L257: Fig2 legend “Experiments for CoV, EcV, and ENK groups were performed in mice without vagotomy. Experiments for vCoV, vEcV, and vENK groups were in mice with vagotomy. Data values were indicated as mean ± SD (n = 6). Means with same letters are not significantly different (p < 0.05). “Is it about graph L)-M)? The author should describe in Fig 2 legends
-->Thank you. We revised them.
L401 induced colitis and gut dysbiosis. tNK109 and its lysates NC and NM also alleviated cognitive. What is tNK109? Should delete “t”, should not?
-->Thank you. We revised them (t means ‘tyndallized’) in Line 76.
Round 2
Reviewer 1 Report
In this manuscript, the authors described the important role exerted by the Lactobacillus gasseri NK109 in alleviating neuropsychiatric disorders and colitis by modulating IL-1b expression, gut microbiota, and vagus nerve-mediated gut-brain signaling.
I have one question regarding the dose used for the Lactobacillus gasseri NK109. Based on what, did you decided to use this concentration (109 CFU/mouse/day). Are you able to observe these effects only at a high concentration?
Author Response
In this manuscript, the authors described the important role exerted by the Lactobacillus gasseri NK109 in alleviating neuropsychiatric disorders and colitis by modulating IL-1b expression, gut microbiota, and vagus nerve-mediated gut-brain signaling.
I have one question regarding the dose used for the Lactobacillus gasseri NK109. Based on what, did you decided to use this concentration (109 CFU/mouse/day). Are you able to observe these effects only at a high concentration?
--> Thank you for your comment. We orally gavaged NK109 (109 CFU/mouse/day) in mice with K1-induced cognitive impairment. We added it in Figure 3 legend and yellow-highlighted in supplement PDF-file. However, we did not treat it at doses of >109 CFU/mouse/day. We hope to perform it in the further study.